



# Validation of satellite formaldehyde (HCHO) retrievals using observations from 12 aircraft campaigns

Lei Zhu[1,2], Gonzalo González Abad[1], Caroline R. Nowlan[1], Christopher Chan Miller[1], Kelly Chance[1], Eric C. Apel[3], Joshua P. DiGangi[4], Alan Fried[5], Thomas F. Hanisco[6], Rebecca S. Hornbrook[3], Lu Hu[7], Jennifer Kaiser[8,9], Frank N. Keutsch[10,11,12], Wade Permar[7], Jason M. St. Clair[6,13], Glenn M. Wolfe[6,13]

[1]Harvard-Smithsonian Center for Astrophysics, Cambridge, MA, USA
[2]School of Environmental Science and Engineering, Southern University of Science and Technology, Shenzhen, China
[3]Atmospheric Chemistry Observations & Modeling Laboratory, National Center for Atmospheric Research, Boulder, CO, USA
[4]NASA Langley Research Center, Hampton, VA, USA
[5]Institute of Arctic & Alpine Research, University of Colorado, Boulder, CO, USA
[6]Atmospheric Chemistry and Dynamic Laboratory, NASA Goddard Space Flight Center, Greenbelt, MD, USA
[7]Department of Chemistry and Biochemistry, University of Montana, Missoula, MT, USA
[8]School of Civil and Environmental Engineering or Earth, Georgia Institute of Technology, Atlanta, GA, USA
[9]School of Earth and Atmospheric Sciences, Georgia Institute of Technology, Atlanta, GA, USA
[10]John A. Paulson School of Engineering and Applied Sciences, Harvard University, Cambridge, MA 02138, USA
[11]Department of Chemistry and Chemical Biology, Harvard University, Cambridge, MA 02138, USA
[12]Department of Earth and Planetary Sciences, Harvard University, Cambridge, MA 02138, USA
[13]Joint Center for Earth Systems Technology, University of Maryland Baltimore County, Baltimore, MD, 21228, USA

*Correspondence to*: Lei Zhu (lei.zhu.02@gmail.com)

**Abstract.** Formaldehyde (HCHO) has been measured from space for more than two decades. Owing to its short atmospheric lifetime, satellite HCHO data are used widely as a proxy of volatile organic compounds (VOCs; please refer to Appendix A for abbreviations and acronyms), providing constraints on underlying emissions and chemistry. However, satellite HCHO products from different satellite sensors using different algorithms have received little validation so far. The accuracy and consistency of HCHO retrievals remain largely unclear. Here we develop a global validation platform for satellite HCHO retrievals using *in situ* observations from 12 aircraft campaigns with a chemical transport model (GEOS-Chem) as the intercomparison method. Application to the NASA operational OMI HCHO product indicates slight biases (−30.9% to +16.0%) under high-HCHO conditions partially caused by *a priori* shape factors used in the retrievals, while high biases (+113.9% to +194.6%) under low-HCHO conditions due mainly to slant column fitting and radiance reference sector correction. By providing quick assessment to systematic biases in satellite products over large domains, the platform facilitates, in an iterative process, optimization of retrieval settings and the minimization of retrieval biases. It is also complementary to localized validation efforts based on ground observations and aircraft spirals.



## 1 Introduction

Formaldehyde (HCHO) is ubiquitous in the troposphere due to its high product yields from atmospheric oxidation of volatile

organic compounds (VOCs). Methane mainly controls the tropospheric background, whereas regional enhancements are contributed largely by short-lived non-methane VOCs (NMVOCs) emitted from the biosphere, human activities, and wildfires. HCHO is detectable from space using solar ultraviolet backscattered radiation between 325 and 360 nm [Chance et al., 2000]. HCHO vertical column densities (VCDs; in the unit of molecules $cm^{-2}$) are obtained after the retrieval process and the consideration of *a priori* information. Because of the short atmospheric lifetime of HCHO (a few hours), satellite HCHO VCD

has been used as a localized proxy for NMVOC emissions [*e.g.*, Palmer et al., 2003; Shim et al., 2005; Stavrakou et al., 2009; Marais et al., 2012; Barkley et al., 2013; Zhu et al., 2014; Zhu et al., 2017a; Cao et al., 2018; Surl et al., 2018]. In addition, previous applications of HCHO retrievals also include evaluating surface ozone sensitivity [Jin and Holloway, 2015; Jin et al., 2017], quantifying cancer risks of ambient HCHO [Zhu et al., 2017b], estimating organic aerosol abundance [Liao et al., 2019], and mapping hydroxyl (OH) radicals [Wolfe et al., 2019]. However, validation of satellite HCHO products from different

satellite sensors using different algorithms have received little attention so far. Validation exercises over different regions in different seasons remain extremely limited. Here we develop a validation platform built with HCHO observations from 12 aircraft campaigns over the United States, Eastern Asia, and the remote Pacific Ocean. We further apply it to the NASA operational HCHO product and report the validation results.

HCHO has been continuously observed from space for more than two decades since GOME (1996–2003) [Chance et al., 2000; De Smedt et al., 2008] and SCIAMACHY (2003–2012) [Wittrock et al., 2006; De Smedt et al., 2008]. Presently available observations are from OMI (2004–) [De Smedt et al., 2015; González Abad et al., 2015], GOME-2A (2006–) [De Smedt et al., 2012], OMPS (2011–) [Li et al., 2015; González Abad et al., 2016], GOME-2B (2012–) [De Smedt et al., 2012], and TROPOMI (2018–) [De Smedt et al., 2018]. Hourly HCHO observations (in daytime) will be made available from a

constellation of geostationary satellites to be launched in the coming 1–3 years, including GEMS (2020) [Kim et al., 2019; Kwon et al., 2019] over Eastern Asia, TEMPO (2022) [Zoogman et al., 2017] over North America, and Sentinel-4 (2023) [Courrèges-Lacoste et al., 2017] over Europe. HCHO retrieved from the above satellites generally follows a two-step approach, slant column density (SCD) fitting and conversion of it to VCD using localized air mass factors (AMFs), with retrieval errors being introduced in each step [Marais et al., 2012; De Smedt et al., 2015; González Abad et al., 2015; Hewson et al., 2015;

Kwon et al., 2017; Herman et al., 2018; Nowlan et al., 2018].

Previous validation of HCHO satellite data sets is often conducted by directly comparing coincident satellite pixels and observation points. Wittrock et al. [2006] and Vigouroux et al. [2009] found SCIAMACHY HCHO columns are unbiased compared with ground-based measurements over remote regions. De Smedt et al. [2015] reported OMI and GOME2 data are

–20% to –40% biased against observed vertical profiles. Wang et al. [2017] reported biases in OMI and GOME2 data of –12%



to –20% over the Eastern China from May to December. A recent study showed monthly bias in OMI data ranges from –11% in summer to +26% in winter in Beijing between 2010 and 2016 [Wang et al., 2019]. Comparison with aircraft observations indicated that GOME data are +16% biased during summer over Eastern Texas in the United States [Martin et al., 2004], and that OMI data are biased by –37% in October over Guyana [Barkley et al., 2013]. Tan et al. [2018] found OMPS data are –18% biased against ship-based measurements in June over the East China Sea.

Such direct validation approaches, however, face three practical challenges. First, they require the averaging of extensive observations to reduce large random noises associated with individual satellite retrievals. Second, they fail to make full use of precise *in situ* observations. Low earth orbit (LEO) satellites pass over a certain location within a fixed time window up to a couple of times per day, meaning only a small fraction of observations are coincident with satellite pixels thus suitable for the purpose of direct validation. Finally, reliability of validation results is unclear for areas beyond the observation sites/domains.

Alternatively, Zhu et al. [2016] proposed an indirect validation approach with a chemical transport model (CTM) as the intercomparison method. This approach increases considerably the range of data and conditions that can be used for validation, and therefore reduces random noises in satellite retrievals through averaging. Using this approach, Zhu et al. [2016] found current HCHO satellite products are biased by –20% to –51% against the SEAC$^4$RS [Toon et al., 2016] aircraft measurements over the Southeastern United States during the summer of 2013. Here we follow this indirect validation approach to develop a global validation platform for satellite HCHO retrievals using observations from 12 aircraft campaigns all over the world, as discussed below.

## 2 HCHO observations from aircraft campaigns

Figure 1 shows flight tracks of 12 aircraft campaigns used in this study. Detailed information is summarized in Table 1. Together, the 12 aircraft campaigns offer exceptional opportunities for global validating of satellite HCHO retrievals with extensive observations over the United States (C1–C9; DISCOVER-AQ California 2013, NOMADSS, SENEX, DISCOVER-AQ Texas 2013, DISCOVER-AQ Colorado 2014, FRAPPÉ, WINTER, SONGNEX, and WE-CAN, respectively), Eastern Asia (C10; KORUS-AQ), and the remote Pacific Ocean (C11–C12; ATom-1 and ATom-2). The aircraft campaigns have great spatial coverages over HCHO hotspots, such as the Southeastern United States (C2 and C3) dominated by strong biogenic isoprene emissions [Guenther et al., 2012], Houston area (C4) featured with high anthropogenic NMVOCs [Zhu et al., 2014], and the Western United States (C9) influenced by wildfires. The campaigns also survey different seasons of the year, enabling assessment of seasonal biases in satellite HCHO products.

During the aircraft campaigns, HCHO observations were made along the flight tracks with multiple instruments, including (1) NCAR Difference Frequency Generation Absorption Spectrometer (DFGAS) [Weibring et al., 2006, 2007, 2010], (2) Trace



Organic Gas Analyzer (TOGA) [Apel et al., 2003; 2010; 2015], (3) In Situ Airborne Formaldehyde instrument (ISAF) [Cazorla et al., 2015], (4) Compact Atmospheric Multispecies Spectrometer (CAMS) [Fried et al., 2011; Richter et al., 2015], and (5)

Proton Transfer Reaction Time-of-Flight Mass Spectrometer (PTR-ToF-MS) [Müller et al., 2014]. The instrument accuracy (1σ level) is 4.5%, 15% (lower limit; https://airbornescience.nasa.gov/instrument/TOGA), 10% [Cazorla et al., 2015], 4% [Richter et al., 2015], and 60% [Hu and Permar, 2019] for DFGAS, TOGA, ISAF, CAMS, and PTR-ToF-MS, respectively. The corresponding instrument detection limits are 40–100 ppt [Nowlan et al., 2018], 20 ppt [Wofsy et al., 2018], 36 ppt [Cazorla et al., 2015], ~ 40 ppt [Richter et al., 2015] and 300 ppt [Hu and Permar, 2019], respectively.


HCHO observations from different instruments are generally consistent. Zhu et al. [2016] reported ISAF to be in good agreement with CAMS during the SEAC⁴RS campaign with a correlation coefficient ($r$) of 0.99 and a slope of 1.10. ISAF is also found consistent ($r$=0.98) with DFGAS during the DC3 campaign [Barth et al., 2015] with a slope of 1.07 [Liao et al., 2019]. Figure 2 shows point-to-point comparisons among 1-min averaged TOGA, ISAF, and CAMS HCHO observations

aboard the aircrafts. There is a high correlation in the mixed layer (here and elsewhere defined as below 2 km; $r$=0.86) and free troposphere ($>$ 2 km; $r$=0.93) between TOGA and CAMS during the FRAPPÉ campaign with a Reduced major axis (RMA) regression slope of 1.05±0.01. During the WINTER campaign, TOGA generally matches with ISAF ($r$=0.72) within the mixed layer. However, consistency between the two instruments begins to fall apart in the free troposphere ($r$=0.33), which is likely driven by sampling differences. TOGA correlates highly with ISAF during the ATom-2 (C12) campaign in both mixed layer

($r$=0.83) and free troposphere ($r$=0.82), but overall it is 48% higher than ISAF likely due to the fact that the two instruments are independently calibrated. In this study, we use CAMS data for FRAPPÉ (C6), ISAF data for both WINTER (C7) and ATom-2 (C12), given their higher accuracies.

Figure 3 shows mean vertical profiles measured from the 12 aircraft campaigns. For campaigns conducted over/near land (C1–

C10), aircraft observations show higher level of HCHO within the mixed layer as a result of biogenic and anthropogenic NMVOC emissions. In the free troposphere, HCHO starts to drop sharply due to short lifetimes of highly reactive NMVOCs, such as isoprene (~ 1 h) and HCHO itself (~ 2 h). We see enhanced HCHO (~ 2 ppb) in 4–5 km during the WE-CAN (C9) campaign, which is caused by intensive primary and secondary production of HCHO from wildfires in the Western United States. Mean HCHO over the remote Pacific Ocean (C11–C12) declines with altitudes through the troposphere (below 12 km),

suggesting oxidation of well-mixed methane as the dominant source of the tropospheric background HCHO.

**3 GEOS-Chem as the intercomparison method**

The indirect validation approach requires a CTM to bridge sampling gaps between aircraft observations and satellite retrievals [Zhu et al., 2016]. Here we use GEOS-Chem version 12.0.0 (doi:10.5281/zenodo.1343547) as the intercomparison method for validation of satellite HCHO columns using aircraft observations. With a detailed representation of ozone-NO$_x$-VOCs-aerosol-


halogens tropospheric chemistry, the GEOS-Chem model has been used extensively in several studies to simulate HCHO including comparisons with *in situ* observations [Jaeglé et al., 2015; Zhu et al., 2016; Chan Miller et al., 2017; Liao et al., 2019]. Zhu et al. [2016] and Chan Miller et al. [2017] found that GEOS-Chem provides an unbiased simulation of SEAC⁴RS and SENEX aircraft observations within the mixed layer over the Southeastern United States in summer, including horizontal patterns and mean vertical profiles. In winter, GEOS-Chem is biased by −32% compared against aircraft observations below

300 m over the Northeastern United States [Jaeglé et al., 2015].

The GEOS-Chem model is driven by the Goddard Earth Observing System–Forward Processing (GEOS-FP) assimilated meteorological data, produced by the NASA Global Modeling and Assimilation Office (GMAO) [Molod et al., 2012]. The GEOS-FP meteorological data have a native horizontal resolution of 0.25°×0.3125° with 72 vertical pressure levels and 3 h

temporal frequency (1 h for surface variables and mixed layer depths). Biogenic VOC emissions are from the MEGAN 2.1 model [Guenther et al., 2012] as implemented in GEOS-Chem by Hu et al. [2015]. Anthropogenic emissions are based on the NEI2011 inventory [EPA, 2015] over the United States, and the MIX inventory [Li et al., 2017] over the Eastern Asia region. Fire emissions are from the fourth-generation global fire emissions database (GFED4) [Giglio et al., 2013]. Surface-driven vertical mixing up to the mixing depth is based on the non-local mixing scheme of Holtslag and Boville [1993], as implemented

in GEOS-Chem by Lin and McElroy [2010].

We run the GEOS-Chem model at a 2°×2.5° resolution to simulate the ATom-1 (C11) and ATom-2 (C12) campaigns as HCHO over the remote Pacific Ocean is relatively homogeneously distributed due to methane oxidation. Over the continents, we use the native resolution (0.25°×0.3125°, nested version) in GEOS-Chem to better represent heterogeneities in emissions and

chemistry during the aircraft campaigns (C1–C10) over North America (130°–60°W, 9.75°–60°N) and Eastern Asia (70°–140°E, 15°–55°N). Dynamic boundary conditions for the nested simulations are from global 2°×2.5° runs. Global and nested simulations are spun up for 10 and 1 month, respectively, to remove the sensitivity to initial conditions. GEOS-Chem is sampled along the flight tracks at the time and locations of the aircraft measurements.

Figure 3 shows GEOS-Chem mean HCHO profiles. Previous studies [Scarino et al., 2014; Millet et al., 2015; Zhu et al., 2016] found GEOS-FP mixing depth in summer is biased low comparing with observations by a factor of 30%−50%, which may partially contribute to the underestimation of HCHO in the mixed layer (Figure 3) by GEOS-Chem over the Unites States (C1–C6, C9) and South Korea (C10). On top of that, underestimation of highly reactive VOC emissions as reported by Zhu et al. [2014] may be another reason for the lower simulated HCHO over Houston area (C4). GEOS-Chem generally reproduces the

observed vertical distribution of HCHO in the free troposphere. Exceptions are for campaigns surveying the Western United States in summer (C5, C6, and C9), likely caused by uncertainties in GFED4 fire emissions in the model.





By integrating the mean vertical profiles in Figure 3, we estimate, for each aircraft campaign, a mean observed HCHO column, a mean GEOS-Chem modeled HCHO column, and the regional bias associated with GEOS-Chem model as informed by

comparison between observed and modelled HCHO columns. Figure 3 shows the regional bias for each aircraft campaign, which is later applied as the correction factor in the validation exercises.

## 4 Application to NASA operational HCHO product

NASA operational OMI HCHO product is based on the Smithsonian Astrophysical Observatory (SAO) HCHO retrieval algorithm [González Abad et al., 2015]. Briefly, the algorithm follows a two-step approach. First, a radiance reference sector

correction term ($\Omega_{S0}$) is subtracted from the fitted total SCD ($\Omega_S$), yielding the radiance reference sector corrected SCD ($\Delta\Omega_S$):

$$\Delta\Omega_S = \Omega_S - \Omega_{S0} \tag{1}$$

Following Khokhar et al. (2005) and De Smedt et al. (2008), the radiance reference sector correction ($\Omega_{S0}$) represents a daily post-processing normalization for the retrieved SCD, calculated as the difference between the retrieved SCD over the Pacific Ocean and the GEOS-Chem climatology (González Abad et al., 2015; González Abad et al., 2016). $\Delta\Omega_S$ is then converted to

VCD ($\Omega$) by applying the localized air mass factor ($AMF$):

$$\Omega = \frac{\Delta\Omega_S}{AMF} \tag{2}$$

The $AMF$ depends on a number of factors, including solar zenith angle ($\theta_Z$), satellite viewing angle ($\theta_V$), cloud characteristics, scattering properties of the atmosphere and surface, and HCHO *a priori* profiles. Following Palmer et al. [2001], it is computed as the product of a geometrical $AMF$ ($AMF_G$) and a correction with scattering weights $w$ applied to the vertical shape factors

$S$:

$$AMF = AMF_G \int_{P_S}^{0} w(p)S(p)dp \tag{3}$$

$$AMF_G = \sec\theta_Z + \sec\theta_V \tag{4}$$

Here the integration is over the pressure ($p$) coordinate from the surface ($P_S$) to the top of atmosphere. $S$ is the normalized vertical profile of HCHO mixing ratios $C(p)$:

$$S(p) = \frac{C(p)\Omega_A(p)}{\int_{P_S}^{0} C(p)\Omega_A(p)dp} \tag{5}$$

where $\Omega_A(p)$ is the partial air column density at $p$, and $w$ measure the sensitivity of the backscattered radiation to HCHO. OMI SAO HCHO product provides $\Omega$, $\Omega_S$, $\Delta\Omega_S$, $AMF_G$ (in term of $\theta_Z$ and $\theta_V$), $AMF$, $S$, and $w$ for each pixel. Uncertainties associated with $\Omega$ are 45–105%, contributed by uncertainties in both $AMF$ (~ 35%) and $\Delta\Omega_S$ (30–100%) [González Abad et al. 2015].

Here we use the DISCOVER-AQ 2013 (C1) flight campaign as an example to demonstrate the validation process. Validation of OMI SAO HCHO product starts with the selection of satellite pixels. This is done for each campaign within the corresponding study period (Table 1) and domain (defined in Table 1; shown in Figure 1) based on following criteria: (1) pass



quality checks (MainDataQualityFlag=0), (2) have cloud fraction less than 0.3, (3) have $\theta_Z$ less than 60°, and (4) have VCD

within the range of $-0.5\times10^{16}$ molecules cm$^{-2}$ to $1.0\times10^{17}$ molecules cm$^{-2}$. We then compute campaign-averaged GEOS-Chem

HCHO columns by sampling the model according to OMI's schedule. The original GEOS-Chem columns are further scaled

using correction factors informed by comparison of model and aircraft columns (Figure 3). Figure 4 shows campaign-averaged

HCHO columns for both OMI SAO and corrected GEOS-Chem over the study domain (California, United States). Campaign-

averaged OMI and corrected GEOS-Chem HCHO columns for other campaigns (C2–C12) are in the Supplement. Poor spatial

correlations between OMI and corrected GEOS-Chem columns during some campaigns (Figure 4 and Supplement) likely

reflect large uncertainties in OMI columns. Finally, we compare spatially and temporally averaged HCHO columns, during

the study period and over the study domain, as reported by OMI SAO product and modelled by GEOS-Chem (with correction)

to estimate the regional systematic bias in OMI SAO HCHO product. Detailed validation results are summarized in Table 2.

We see from Table 2 that relative biases in OMI HCHO product depends on both locations and seasons, ranging from –30.9%

over South Korea in summer (C10) to +194.6% over Western United States in spring (C8). Overall, the relative biases in OMI

SAO product fall into two categories. First, the product is slightly biased (–30.9% to +16.0%) under high-HCHO conditions

(defined as mean HCHO column > $1.10\times10^{16}$ molecules cm$^{-2}$), such as summertime Southeastern United States (C2, C3, C4)

and summertime South Korea (C10). A similar bias (–37.0%) in OMI SAO HCHO product is reported by Zhu et al. [2016] for

summertime Southeastern United States. Second, the product is highly biased (+113.9% to +194.6%) under low-HCHO

conditions, such as the Western United States (C5, C6, and C8), wintertime United States (C1 and C7), and the remote Pacific

Ocean (C11 and C12). Our work points to a higher bias (~ 120%) in OMI SAO retrievals over the remote Pacific Ocean

compared with the bias (~ 10%) reported by Wolfe et al. [2019]. This is likely driven by a number of factors: (1) Wolfe et al.

[2019] use all data, whereas we only use data over the Pacific region (Figure 1); (2) radiance reference sector correction is

treated differently in the two studies; (3) selection criteria for OMI pixels are different; (4) mean observed HCHO column is

computed from individual profiles in Wolfe et al. [2019], while it is computed based on a mean profile in this study; (5) and

finally, the relative bias metric is more sensitive to absolute bias under low-HCHO conditions.

We attribute biases in the first case partially to *a priori* vertical profiles used in the SAO HCHO retrieval algorithm, in particular

underestimate of HCHO in the mixed layer. SAO HCHO algorithm samples HCHO shape factors ($S$) from a monthly mean

climatology based on GEOS-Chem simulations in 2007 at a spatial resolution of 2°×2.5°, which may not be able to represent

the spatial heterogeneity in chemistry, nor/or to model temporal variations in emissions. After recomputing the AMF with

observed HCHO shape factors following equation (3)–(5), relative biases in HCHO can be reduced on average from –15.9%

to –8.4% (C3, C4, C9, and C10 in Table 2). As shown in Figure 5, using observed HCHO shape factors (C3, C4, and C9)

results in lower AMF by correcting underestimated *a priori* HCHO within the mixed layer. During the WE-CAN campaign





(C9), recomputed AMF is slightly higher than that reported by OMI (Table 2) because of elevated HCHO around 3–5 km from wildfire plumes (Figure 5).

In the second case, using observed HCHO shape factors, however, barely reduces biases in OMI SAO HCHO product (Table 2), implying that radiance reference sector corrected SCD ($\Delta\Omega_S = \Omega_S - \Omega_{S0}$) rather than AMF is likely the main driver of high

biases. This can be further examined with aircraft observations and OMI HCHO pixels over remote Pacific Ocean (C11 and C12), where contribution of $\Omega_{S0}$ to $\Delta\Omega_S$ is much lower (~ 15%; Table 2). Integration of ATom1 (C11) and ATom2 (C12) vertical profiles indicates a Pacific background HCHO VCD of ~ $3.0 \times 10^{15}$ molecules cm$^{-2}$ (Figure 3), comparable with previous measured values ($2.8 \times 10^{15}$ molecules cm$^{-2}$ to $4.6 \times 10^{15}$ molecules cm$^{-2}$) over the remote North Pacific Ocean [Singh et al., 2009] and modeled results ($4.5 \times 10^{15}$ molecules cm$^{-2}$) [Wolfe et al., 2019]. This is equivalent to a background SCD of ~

$4.7 \times 10^{15}$ molecules cm$^{-2}$ with AMF computed using observed HCHO shape factors (Figure 5). OMI SAO SCD ($\Omega_S$) and radiance reference sector corrected SCD ($\Delta\Omega_S$) is much higher than such estimated background SCD value by a factor of 2.0 to 2.5 (Table 2), pointing to potential issues with SCD fitting and/or radiance reference sector correction in the SAO HCHO retrieval algorithm.

The SAO retrieval algorithm conducts the radiance reference sector correction by removing the contribution of HCHO over the remote Pacific Ocean to the radiance reference. This HCHO contribution is derived using a high-resolution solar spectrum [Chance et al., 2010] convolved with the instrument response function. Despite OMI's stability over the mission lifetime [Schenkeveld et al., 2017], small spectral changes could have significant impacts on the derived HCHO columns over the remote Pacific Ocean where HCHO signals are relatively weak. We revaluate such impacts by supressing removal of HCHO

contribution in the radiance reference. The new approach improves both spectral fitting results and retrieval stability during the life span of OMI. In consequence, mean bias in the resulted columns is reduced from 147.1% (Table 2) to 128.2% in the second case. We attribute the remaining biases to (1) increased impact of interferers (*e.g.*, $O_3$ and BrO, $O_2$-$O_2$ and water vapor) when HCHO signals are weak and (2) the latitudinal dependency of the radiance reference sector correction. We also find that OMI SAO HCHO VCD correlates moderately ($r$=0.38 to 0.66) with surface albedo during some campaigns (C1, C5, and C6),

suggesting possible bias introduced by using a reflectance climatology [Kleipool et al., 2008] in the retrievals. In summary, high biases under low-HCHO conditions are likely driven by both radiance reference sector correction and SCD fitting. An updated SAO product is being developed to minimize the biases by optimizing the two processes accordingly.

## 5 Conclusions

We have used HCHO observations from 12 aircraft campaigns, together with the GEOS-Chem chemical transport model as

an intercomparison method, to develop a global validation platform for satellite HCHO retrievals. The global validation platform offers an alternative way to quickly assess systematic biases in satellite products over large spatial domains and longer





temporal periods, facilitating optimization of retrieval settings and the minimization of retrieval biases. Application to NASA operational HCHO product (SAO retrievals) indicates that relative biases range from –30.9% to +194.6% depending on locations and seasons. Under high-HCHO conditions, such as summertime Southeastern United States, the product is slightly

biased (–30.9% to +16.0%) due partially to underestimate of HCHO within the mixed layer by *a priori* profiles. Under low-HCHO conditions, such as wintertime United States and remote Pacific Ocean, the product is highly biased (+113.9% to +194.6%), likely as a result of slant column density fitting process of HCHO. Our work points to the need for improvement in OMI SAO HCHO product to correct the systematic biases, particularly, optimization of the HCHO slant column fitting and reference sector correction.

**Data and code availability**

The validation platform (R scripts) is available at: https://doi.org/10.7910/DVN/KG3XNC.

The GEOS-Chem model is available at http://acmg.seas.harvard.edu/geos/ (last access: Nov. 29, 2019).

OMI-SAO HCHO data were downloaded from http://disc.sci.gsfc.nasa.gov/Aura/dataholdings/OMI/omhcho_v003.shtml.

Aircraft observations are available respectively as following:

DISCOVER-AQ California 2013 (C1): https://www-air.larc.nasa.gov/missions/discover-aq/discover-aq.html

NOMADSS (C2): https://www.eol.ucar.edu/field_projects/nomadss/

SENEX (C3): https://www.esrl.noaa.gov/csd/projects/senex/

DISCOVER-AQ Texas 2013 (C4): https://www-air.larc.nasa.gov/missions/discover-aq/discover-aq.html

DISCOVER-AQ Colorado 2014 (C5): https://www-air.larc.nasa.gov/missions/discover-aq/discover-aq.html

FRAPPÉ (C6): http://catalog.eol.ucar.edu/FRAPPE/

WINTER (C7): http://catalog.eol.ucar.edu/winter/

SONGNEX (C8): https://www.esrl.noaa.gov/csd/projects/songnex/

WE-CAN (C9): https://www.eol.ucar.edu/field_projects/we-can/

KORUS-AQ (C10): https://www-air.larc.nasa.gov/missions/korus-aq/

ATom-1 (C11): https://daac.ornl.gov/ATOM/campaign/

ATom-2 (C12): https://daac.ornl.gov/ATOM/campaign/

**Appendix A**

*Abbreviations and acronyms*

| AMF | Air mass factor |
|---|---|
| 285 AMF$_G$ | Geometrical Air mass factor |
| ATom | Atmospheric Tomography Mission |



| | | |
|---|---|---|
| | CAMS | Compact Atmospheric Multispecies Spectrometer |
| | CTM | Chemical transport model |
| | DC3 | Deep Convective Clouds and Chemistry Experiment |
| 290 | DFGAS | Difference Frequency Generation Absorption Spectrometer |
| | DISCOVER-AQ | Deriving Information on Surface Conditions from COlumn and VERtically Resolved Observations Relevant to Air Quality |
| | FRAPPÉ | Front Range Air Pollution and Photochemistry Éxperiment |
| | GEMS | Geostationary Environment Monitoring Spectrometer |
| 295 | GEOS-FP | Goddard Earth Observing System–Forward Processing |
| | GFED4 | Fourth-generation Global Fire Emissions Database |
| | GMAO | Global Modeling and Assimilation Office |
| | GOME(-2) | Global Ozone Monitoring Experiment(-2) |
| | HCHO | Formaldehyde |
| 300 | ISAF | In Situ Airborne Formaldehyde |
| | KORUS-AQ | Korea-United States Air Quality |
| | LEO | Low Earth Orbit |
| | MEGAN | Model of Emissions of Gases and Aerosols from Nature |
| | NEI | National Emissions Inventory |
| 305 | NMVOCs | Non-Methane VOCs |
| | NOMADSS | Nitrogen, Oxidants, Mercury, and Aerosol Distributions, Sources, and Sinks |
| | OMI | Ozone Monitoring Instrument |
| | OMPS | Ozone Mapping and Profiler Suite |
| | PTR-ToF-MS | Proton-Transfer-Reaction Time-of-Flight Mass Spectrometer |
| 310 | RMA | Reduced major axis |
| | SAO | (Harvard) Smithsonian Astrophysical Observatory |
| | SCD | Slant Column Density |
| | SCIAMACHY | Scanning Imaging Absorption spectroMeter for Atmospheric Chartography |
| | SEAC⁴RS | Studies of Emissions, Atmospheric Composition, Clouds and Climate Coupling by Regional Surveys |
| 315 | SENEX | Southeast Nexus |
| | SONGNEX | Shale Oil and Natural Gas Nexus |
| | TEMPO | Tropospheric Emissions: Monitoring of Pollution |
| | TOGA | Trace Organic Gas Analyzer |
| | TROPOMI | TROPOspheric Monitoring Instrument |
| 320 | VCD | Vertical Column Density |



| | |
|---|---|
| VOCs | Volatile Organic Compounds |
| WE-CAN | Western wildfire Experiment for Cloud chemistry, Aerosol absorption and Nitrogen |
| WINTER | The Wintertime INvestigation of Transport, Emissions, and Reactivity |

## Acknowledgements

We acknowledge contributions from science teams of the 12 aircraft campaigns. This work is funded by NOAA Atmospheric Chemistry Carbon Cycle and Climate NA18OAR4310108, NASA Aura Science Team NNX17AH47G, NASA Science of TERRA, AQUA, and SUOMI NPP 80NSSC18K0691, and NASA Making Earth System Data Records for Use in Research Environments 80NSSC18M0091 grants. LZ thanks supports from the Smithsonian Astrophysical Observatory (SAO) Visiting Scientist Fellowship. The 2018 WE-CAN campaign was supported by the National Science Foundation (Grant NSF AGS-
1650275, -1650786, -1650288, -1650493, -1652688). LH and WP would like to acknowledge operational, technical and scientific support provided by NCAR's Earth Observing Laboratory, sponsored by the National Science Foundation. This material is based upon work supported by the National Center for Atmospheric Research, which is a major facility sponsored by the National Science Foundation under Cooperative Agreement No. 1852977. The NASA Goddard Space Flight Center (GSFC) team acknowledges support for the ATom campaign from the NASA Earth Venture Suborbital-2 Program, and support
for DC3 and SEAC$^4$RS campaigns from NASA.

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

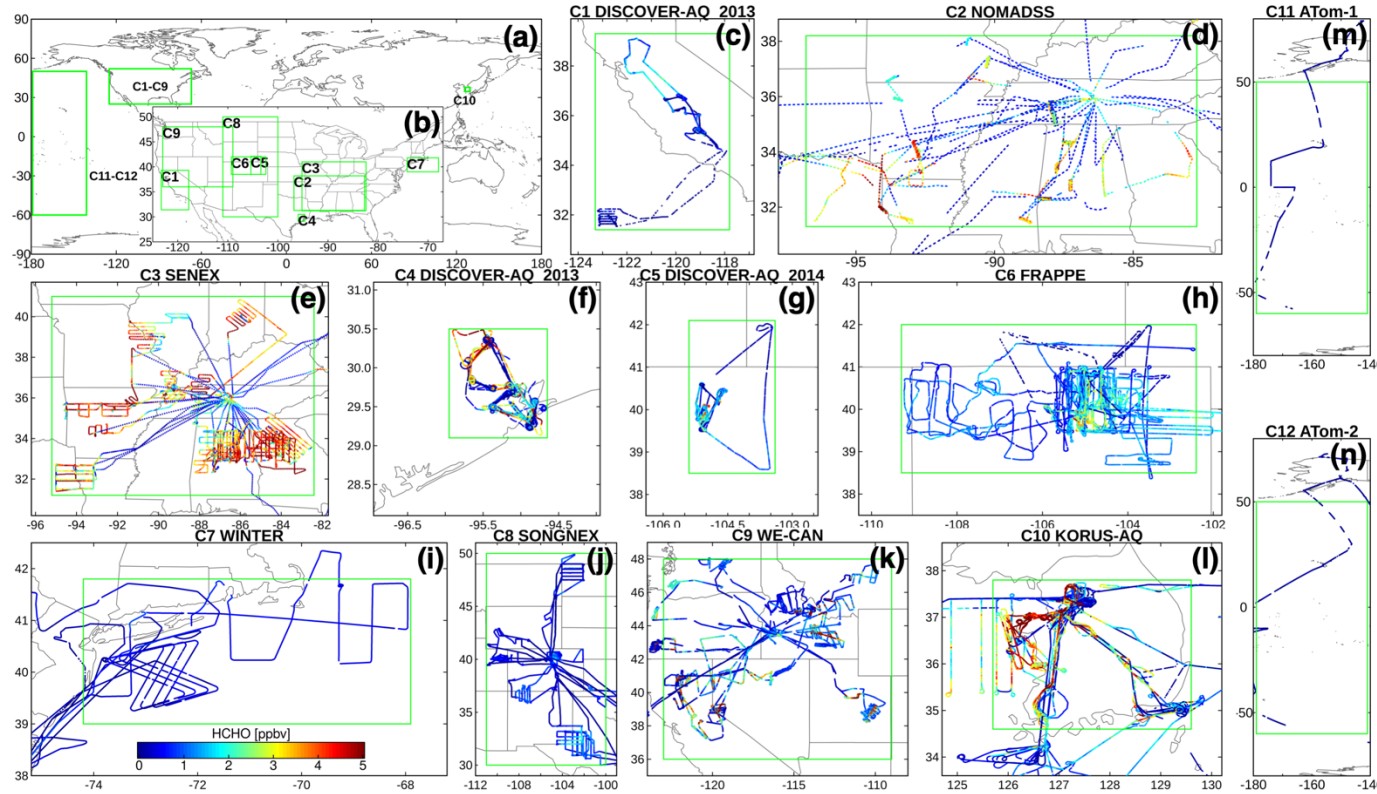

630

**Figure 1.** Flight tracks of the 12 aircraft campaigns used in this study. Panel (a) shows the spatial coverage (green rectangles) of the campaigns. Panel (b) (inset) zooms in campaigns over the United States. Aircraft campaigns are numbered as C1–C12. Table 1 summarizes detailed information of the 12 campaigns. Formaldehyde (HCHO) mixing ratios along aircraft flight tracks are shown in panel (c)–(n). Color bar saturates at 5 ppbV. The green rectangle in panel (c)–(n) is the same as that in (a) and (b), indicating spatial domain of a certain campaign.
635 The same domain is also defined in Table 1.

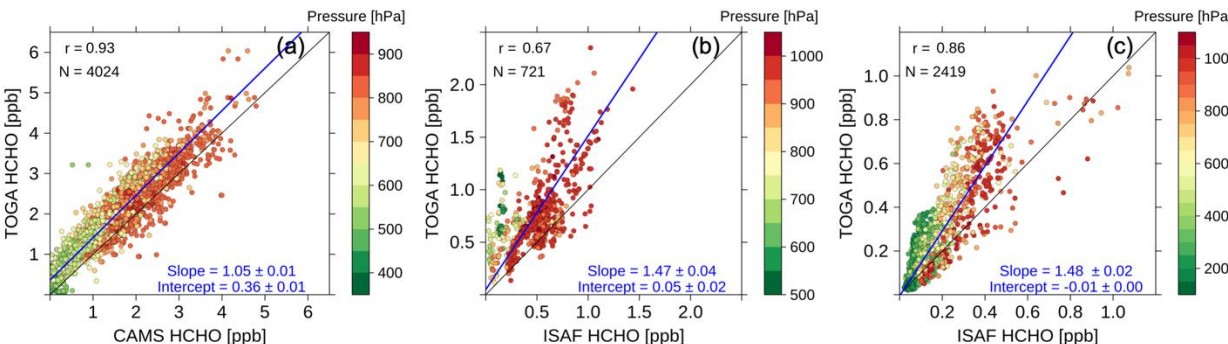

**Figure 2.** Comparisons between 1-min averaged HCHO observations from multiple instruments. (a) Observations from TOGA and CAMS instruments aboard the NSF/NCAR C-130 during the FRAPPÉ campaign. (b) Observations from TOGA and ISAF instruments aboard the NSF/NCAR C-130 during the WINTER campaign. (c) Observations from TOGA and ISAF instruments aboard the NASA DC-8 during the ATom-2 campaign. HCHO data points are colored by atmospheric pressure. Reduced major axis (RMA) regression slops and intercepts are shown along with the correlation coefficient (*r*), sample size (*N*), RMA regression line (blue) and 1:1 line (black).



**Figure 3.** Mean HCHO vertical profiles as observed during the 12 aircraft campaigns (Table 1) and simulated by GEOS-Chem. GEOS-Chem is sampled along the flight tracks at the time and locations of the measurements. We only use observed and modeled HCHO values within the study area, defined by the green rectangle for each campaign in Figure 1. HCHO values are vertically binned in increments of 500 m. Shading gives the standard deviation in the observations. Observed (black) and modeled (red) HCHO column densities ($10^{15}$ molecules cm$^{-2}$) are insert along with relative biases (in parentheses) in modeled column densities. The relative biases are further used as correction factors for GOES-Chem columns. Observed column densities are computed using mean observed mixing ratio (black lines), temperature, and pressure. Modeled column densities are computed according to GEOS-Chem HCHO vertical profiles (red lines) as well as temperature and pressure from GEOS-FP. Notice that scale in panel (k) and (l) is different from that in other panels.



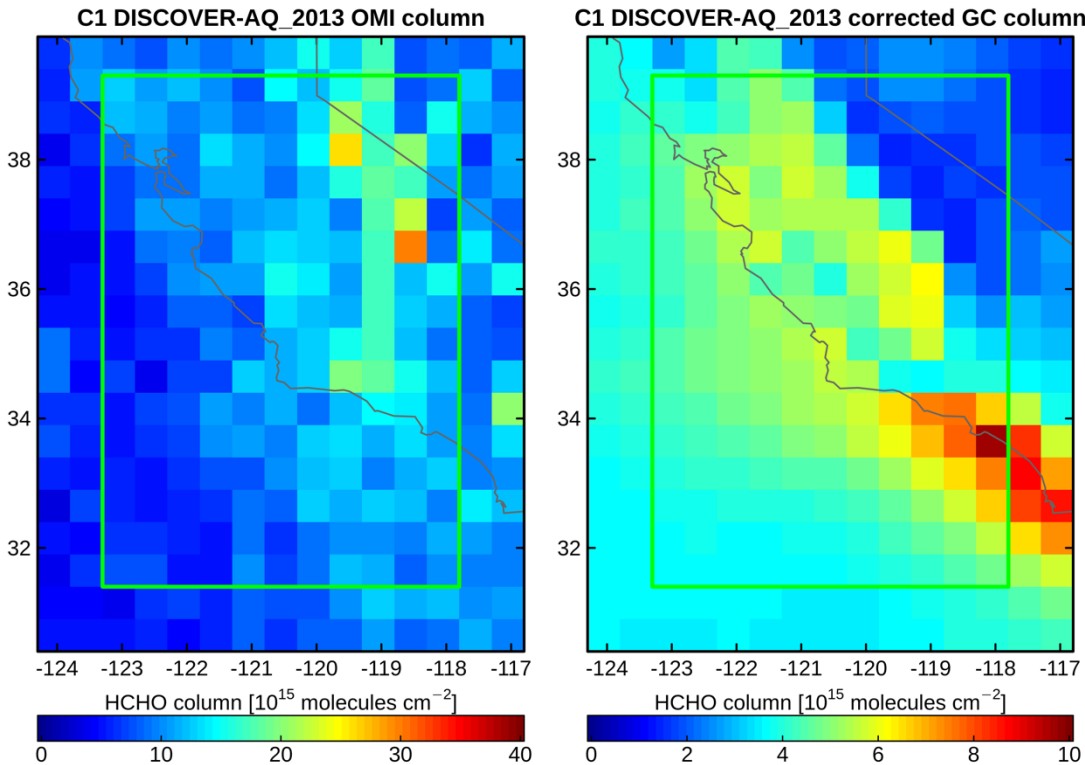

**Figure 4.** HCHO vertical column densities over California, United States during the DISCOVER-AQ California 2013 (C1; 16 Jan. – 6 Feb.). The left panel shows data from OMI SAO HCHO product. The right panel shows GEOS-Chem model results sampled on the OMI schedule (see text), and scaled by a factor of 1.53 to correct for the bias relative to aircraft measurements (Figure 3). OMI and GEOS-Chem results are regridded onto the 0.5°×0.5° grids. The green rectangles represent the study domain (same as that in Figure 1), which is also defined in Table 1. Notice the two panels are in different HCHO scales.

655





**Figure 5.** Air mass factor (AMF) components over the 12 aircraft campaigns. Each panel shows mean scattering weights ($w$; blue dashed line) and shape factors ($S$; blue solid line) from OMI SAO HCHO product averaged over the corresponding study domain (shown in Figure 1; defined in Table 1) during the campaign period (defined in Table 1), as well as the product of the two (blue dotted line) from which mean AMF is derived by vertical integration using equation (4). Each panel also shows observed HCHO shape factors (black solid) from the mean HCHO profile in Figure 3. We use mean HCHO profiles from ATom-1 and ATom-2 (Figure 3) to fill observations above 6 km. Also shown is the product (black dotted line) of mean OMI scattering weights (blue dashed line) and observed HCHO shape factor (black solid). Mean AMF values are given in the legend computed using OMI (blue) and observed (black) shape factors.



**Table 1.** Overview of the 12 aircraft campaigns used in this study

| Campaign ID | Campaign name | Region | Date | Platform | HCHO instrument(s) [a] | Domain [b] | References [c] |
|---|---|---|---|---|---|---|---|
| C1 | DISCOVER-AQ California 2013 | California, U.S. | Jan. 16–Feb. 06 2013 | NASA P-3B | DFGAS (4%) | 31.4–39.3N 123.3–117.8W | 1, 2 |
| C2 | NOMADSS | Southeastern U.S. | Jun. 03–Jul. 14 2013 | NSF/NCAR C130 | TOGA (15%) [d] | 31.3–38.2N 96.8–82.7W | 3 |
| C3 | SENEX | Southeastern U.S. | Jun. 03–Jul. 10 2013 | NOAA WP-3D | ISAF (10%) | 31.2–41.0N 95.2–82.4W | 4 |
| C4 | DISCOVER-AQ Texas 2013 | Texas, U.S. | Sep. 04–Sep. 29 2013 | NASA P-3B | DFGAS (4%) | 29.1–30.5N 95.95–94.65W | 1, 2 |
| C5 | DISCOVER-AQ Colorado 2014 | Colorado, U.S. | Jul. 17–Aug. 10 2014 | NASA P-3B | DFGAS (4%) | 38.5–42.1N 105.4–103.4W | 1, 2 |
| C6 | FRAPPÉ | Colorado, U.S. | Jul. 26–Aug. 18 2014 | NSF/NCAR C130 | CAMS (4%) TOGA (15%) [d] | 38.5–42.0N 109.3–102.4W | 5, 6 |
| C7 | WINTER | Northeastern U.S. | Feb. 03–Mar. 13 2015 | NSF/NCAR C-130 | ISAF (10%) TOGA (15%) [d] | 39.0–41.8N 72.2–67.9W | 7 |
| C8 | SONGNEX | Western U.S. | Mar. 19–Apr. 27 2015 | NOAA WP-3D | ISAF (10%) | 30.0–50.0N 111.0–100.0W | 8 |
| C9 | WE-CAN | Western U.S. | Jul. 26–Sep. 13 2018 | NSF/NCAR C-130 | PTR-ToF-MS (60%) | 36.0–48.0N 123.0–109.0W | 9, 10 |
| C10 | KORUS-AQ | South Korea | Apr. 26–Jun. 18 2016 | NASA DC-8 | CAMS (4%) | 34.6–37.8N 125.7–129.6W | 11 |
| C11 | ATom-1 | Pacific Ocean | Jul. 29–Aug. 23 2016 | NASA DC-8 | ISAF (10%) | 60.0S–50N 179.0–141.0W | 12 |
| C12 | ATom-2 | Pacific Ocean | Jan. 26–Feb. 21 2017 | NASA DC-8 | ISAF (10%) TOGA (15%) [d] | 60.0S–50N 179.0–141.0W | 12 |

[a] Instrument accuracy is given in parentheses. During C6, C7, and C12, HCHO is measured by two independent instruments
[b] Shown as green rectangles in Figure 1
[c] 1 Crawford and Pickering [2014]; 2 DISCOVER-AQ Science Team [2014]; 3 Emmons [2016]; 4 Warneke et al. [2016]; 5 Pfister et al.
670 [2017]; 6 Richter et al. [2015]; 7 UCAR/NCAR - Earth Observing Laboratory et al. [2016]; 8 National Oceanic and Atmospheric
Administration (NOAA) [2017]; 9 Pollack et al. [2019]; 10 Hu and Permar [2019]; 11 KORUS-AQ [2016]; 12 Wofsy et al. [2018]
[d] TOGA has an accuracy of 15% or better





**Table 2.** HCHO columns and validation results over the 12 aircraft campaigns [a]

| Campaign ID | GEOS-Chem columns | | OMI | | | | | | with observed shape factors | |
| --- | --- | --- | --- | --- | --- | --- | --- | --- | --- | --- |
| | Original [b] | Corrected [c] | $AMF_G$ | AMF | $\Omega_S$ [d] | $\Omega_{S0}$ [e] | $\Omega_{avg}$ [f] | $\Omega_{comp}$ [g] | AMF [h] | $\Omega_{comp}$ [i] |
| C1 | 2.82 | 4.30 | 3.14 | 1.25 | 16.40 | 5.03 | 10.25 (+138.4%) | 9.07 (+110.9%) | 1.07 | 10.66 (+148.0%) |
| C2 | 15.49 | 18.36 | 2.54 | 0.97 | 16.78 | 4.73 | 13.24 (−27.9%) | 12.42 (−32.4%) | 0.96 | 12.58 (−31.5%) |
| C3 | 14.53 | 17.68 | 2.48 | 0.95 | 16.87 | 4.53 | 13.88 (−21.5%) | 12.95 (−26.8%) | 0.83 | 14.85 (−16.0%) |
| C4 | 16.05 | 17.32 | 2.68 | 1.07 | 17.45 | 5.10 | 11.97 (−30.90%) | 11.52 (−33.5%) | 0.88 | 14.06 (−18.8%) |
| C5 | 6.13 | 5.05 | 2.49 | 1.02 | 15.88 | 3.85 | 13.58 (+168.7%) | 11.85 (+134.6%) | 0.87 | 13.89 (+174.8%) |
| C6 [j] | 5.59 | 5.37 | 2.54 | 1.05 | 15.47 | 4.02 | 12.44 (+131.6%) | 10.90 (+102.9%) | 0.86 | 13.30 (+147.7%) |
| C7 [k] | 2.66 | 3.44 | 3.19 | 1.45 | 12.09 | 1.23 | 8.79 (+155.6%) | 7.49 (+117.8%) | 1.48 | 7.33 (+113.1%) |
| C8 | 2.75 | 3.71 | 2.59 | 1.19 | 15.19 | 3.27 | 10.92 (+194.6%) | 10.02 (+170.4%) | 1.26 | 9.47 (+155.5%) |
| C9 | 5.85 | 10.92 | 2.60 | 1.09 | 16.11 | 3.57 | 12.67 (+16.0%) | 11.49 (+5.2%) | 1.19 | 10.55 (−3.5%) |
| C10 | 7.34 | 10.69 | 2.59 | 1.15 | 16.36 | 5.14 | 10.49 (−1.8%) | 9.79 (−8.4%) | 1.07 | 10.45 (−2.3%) |
| C11 | 2.66 | 2.97 | 2.75 | 1.61 | 11.86 | 1.56 | 6.72 (+126.6%) | 6.39 (+115.4%) | 1.56 | 6.60 (+122.5%) |
| C12 [k] | 2.61 | 3.19 | 2.78 | 1.64 | 12.06 | 1.52 | 6.82 (+113.9%) | 6.42 (+101.4%) | 1.61 | 6.53 (+104.8%) |
| SEAC$^4$RS [l] | 15.23 | 16.90 | 2.66 | 0.95 | - | - | 10.60 (−37.0%) | - | - | - |

[a] Results are spatially and temporally averaged values for the study regions (shown as green rectangles in Figure 1 and defined in Table 1) during the study periods (defined in Table 1). HCHO columns (GEOS-Chem columns, $\Omega_S$, $\Omega_{S0}$, $\Omega_{avg}$, and $\Omega_{comp}$) are in the unit of $10^{15}$ molecules cm$^{-2}$. For each aircraft campaign, biases relative to the corrected GEOS-Chem column are given in parentheses

[b] sampled from the GEOS-Chem models according to OMI's schedule

[c] corrected with the factors informed by comparison of observed and modeled HCHO columns (Figure 3)

[d] SCD computed using vertical column density without reference sector correction ("ColumnAmount" data field in OMI SAO HCHO product) and air mass factor (AMF)

[e] SCD correction term recomputed using averaged OMI $\Omega_S$, $\Omega_{avg}$, and AMF following equation (1)

[f] mean VCD by directly averaging valid satellite pixels



[g] VCD recomputed using averaged OMI $\Omega_S$, $\Omega_{S0}$, and AMF following equation (1)

[h] recomputed using averaged OMI $AMF_G$, observed mean HCHO shape factors (Figure 5), and mean OMI scattering weights (Figure 5) following equation (3)–(5)

[i] VCD computed using recomputed AMF, averaged OMI $\Omega_S$, and averaged OMI $\Omega_{S0}$ following equation (1)–(2)

[j] using CAMS observations

[k] using ISAF observations

[l] results reported by Zhu et al. [2016] over the southeastern United States (30.5–39.0N, 95.0–81.5W) during Aug. 05–Sep. 25, 2013. Results are based on a different version of GEOS-Chem model