# Peer review of "Validation of satellite formaldehyde (HCHO) retrievals using observations from 12 aircraft campaigns"

_Atmospheric Chemistry and Physics, 2019_

## Referee Comment (RC1) · Anonymous Referee #1 · 8 Feb 2020

In this paper, Zhu et al. evaluated OMI HCHO retrievals using in situ measurements from 12 airborne field campaigns (mostly over the U.S.) and the GEOS-Chem chemical transport model. They first compared GEOS-Chem simulated HCHO with aircraft measurements and used a constant scaling factor for each field campaign to correct the biases in GEOS-Chem simulations. The bias-corrected GEOS-Chem HCHO vertical column densities (VCDs) were then compared with OMI retrievals. It was found that OMI retrievals were generally biased high for low-HCHO conditions and had relatively small biases for high-HCHO cases. Potential reasons for the biases in OMI retrievals were also discussed. Overall, this is a useful study for understanding the quality and uncertainty in OMI HCHO retrievals. The authors expanded upon their previous study

using the same approach (Zhu et al., 2016) to different locations and seasons. The advantage of the approach is that by using GEOS-Chem as a transfer standard, data from different field campaigns (often involving different instruments) can be used in a more consistent way for satellite validation. The paper is generally well written. There are, however, several points that need to be clarified. I would recommend that the paper be accepted for publication after the following comments have been addressed.

Specific Comments:

González Abad et al. (2015) showed that there were fairly large changes in OMI HCHO retrievals over remote background areas between the beginning of OMI (2004-2005) and more recent years (2012 and later). All the field campaigns discussed in this study took place after 2013. Have the authors looked into data from earlier field campaigns to check , for example, whether OMI retrieval biases under low-HCHO conditions were smaller in earlier years?

Given the positive biases over low-HCHO areas and (smaller but mostly negative) biases over high HCHO areas in GEOS-Chem simulations, using a fixed scaling factor may not be appropriate for some of the field campaigns. For example, as shown in Figure 4, the mean HCHO VCD can range from $\sim 3 * 10^{15}$ molecules/cm$^2$ to over $10^{16}$ molecules/cm$^2$ and it is possible that GEOS-Chem overestimates HCHO over relatively clean areas in the southwest corner of the domain, and at the meantime underestimates HCHO over the southeast corner of the domain. Also by changing the domain slightly (for example, expand the domain to the west), the comparisons between OMI and bias-corrected GEOS-Chem can probably yield somewhat different results.

The authors proposed the validation method as a "global validation platform". Since the field campaigns discussed in this paper had no coverage outside of North America and Asia Pacific region, this is hardly "global". Also for the method to be a "platform", there should be some utilities or applications that facilitate the validation of not just

NASA OMI HCHO retrievals, but also retrievals from other satellite sensors/algorithms. Maybe there should also be functionality to ingest additional field campaign data and/or CTM simulations to be used in validation. I wonder if the authors can discuss these aspects of the validation platform (probably as an appendix).

Technical Comments:

Figure 3 is discussed quite extensively in two parts: lines 119-125 and lines 155-161. Maybe the authors can re-organize and consolidate the two parts?
* * *

---

## Referee Comment (RC2) · Anonymous Referee #2 · 10 Feb 2020

Zhu et al. present a new validation platform for satellite HCHO products, using different aircraft measurements and the model GEOS-Chem as the inter-comparison method. The model is used to make the link between the localised aircraft vertical measurements and the global satellite vertical columns. The method was introduced by Zhu et al. (2016). It is now extended to a larger number of aircraft campaigns, covering a broad range of conditions. An extensive evaluation of the NASA operational OMI HCHO satellite product is presented as an application of the platform. The retrieval steps of the satellite product are examined separately in order to explain the differences between satellite and aircraft results. The paper concludes with a slight bias of the OMI satellite product for high HCHO level, and to a high positive bias for low

columns. This study addresses the need for more systematic validation of the satellite products. The use of several aircraft measurements combined with 3D-CTM is pretty new. It allows for direct and indirect validation of the vertical HCHO profiles, which is lacking in the community. A significant amount of work has been dedicated to this goal, and to present the results in an honest and clear way. However, the paper would benefit from some clarifications, especially at the end of the discussion. I recommend publication after the following points have been improved:

The authors claim for a global validation platform. This is supposed to be achieved through the model, but nothing is shown about this extension of the validation beyond the aircraft domains. Can the author add an illustration of this extension? A global map, or a comparison at another location? Alternatively, remain focused on the regions covered by the aircraft campaigns and leave out the reference to the global method. In the abstract, it is stated that the high biases are due to slant column fitting and radiance sector correction. It is not clearly demonstrated in the paper. The last paragraph of section 4 needs to be revised. The explanations are not clear, and conclusions are drawn without showing any results. It does not hold by itself, and needs to be extended (see my comments below). The study look for systematic biases: Pay attention to data selection that can also introduce systematic bias (see my comments below). Does the biases found in this study match the error provided in the satellite product? What about precision? Does the validation results agree with provided satellite product precision?

Introduction p3, line83: "global validation platform . . ... Using observations from 12 aircraft campaigns all over the world". I don't fully agree that this study is global. Results are shown only at the aircraft campaign locations. And those cover mostly US, Pacific and one over Korea. Important emission regions are missing all over the world. I would rather mention here the diversity of the seasonal coverage.

Application to NASA operational HCHO product Line 169: the so called "radiance reference sector correction term s0" is a slant column. Why do you use the term "radiance"? It is confusing. Does it refer to the reference spectra used during the DOAS fit? If yes,

it should be better explained. Line 174: it is said that s0 is the difference between the retrieved SCD over the Pacific Ocean the GEOS-Chem climatology. Please explain if the model vertical columns are directly subtracted from the satellite slant columns (assumption AMF ~~1 over Pacific Ocean, which is not true), or if an air mass factor is used to convert the GEOS-Chem vertical columns into slant columns. Line 194: selection criteria (4): VCD within the range -0.5x1016 molecules cm-2 to 1.0xe1017 molecules cm-2. Please provide the dispersion of the OMI HCHO VCD. If the VCD dispersion is, let's say, 0.8x1016 molecules cm-2, the lower limit might be too strict, removing a significant part of the distribution (negative columns) when the averaged column is close to zero, therefore biasing the averaged column to high values (case "high biases under low-HCHO conditions"). I suggest to test the impact on the comparisons if a more relaxed selection, based on a classical 3approach, is used. Line 207: Is the limit of 1.1 x 1016 for high-HCHO conditions based on satellite or model results? It looks like it is based on satellite columns, which is strange since it is the dataset to be validated, and it presents biases. Line 213: (2) please elaborate on the differences between the "radiance reference sector correction". Same for point (3) "selection criteria". Line 223: Half of the bias in high-HCHO conditions can be attributed to a priori profiles but not the full bias. Please discuss other possible reasons for the remaining bias. Line 229: Second case, low-HCHO conditions: I don't agree with the first conclusion. The a priori profiles are not the unique error source in the AMF calculation. It is not because the use of a more precise profile does not improve the comparison that the error is not due to AMF uncertainties. It could be due to albedo uncertainty; or imperfect cloud correction. However, I agree with the rest of the paragraph. Please reformulate the second line. Line 240: "The SAO retrieval algorithm conducts the radiance reference sector correction by removing the contribution of HCHO over the remote Pacific Ocean to the radiance reference." What are the units of this contribution? A slant column or a radiance? Please explain what you mean by radiance reference. Line 241: Could you give a number for this HCHO contribution? Line 245: What do you mean by "suppressing removal of HCHO contribution in the radiance reference"? Line 246: "the mean

bias is reduced from 147% to 128%". Where does this 128% come from? Furthermore, the bias reduction is not significant. Line 248: "we attribute the remaining biases to (2) the latitudinal dependency of the radiance reference sector correction." Again this needs further explanations. Is there a latitudinal dependency of the slant columns, if yes positive or negative? And is there a latitudinal dependency in the current reference sector correction of the SAO product? Line 249: You find a significant correlation of the satellite HCHO columns with the surface albedo during some campaigns, maybe even larger than the correlation with the model columns. It is interesting. Please specify for which campaigns? If the reflectance climatology contains uncertainties, it will reflect in AMF uncertainties, that will not correct the slant columns for this kind of dependency. Therefore, it is important to know if AMF are used to compute s0. See my comment for line 174. Also, have you tested to remove sun glint scenes from the comparison for the Pacific regions? Figure4 and supplement: spatial correlation between model and satellite column seems rather low. Please add correlations in Table 2. Why not using the same scale for OMI and the model? for example from 0 to 20x1015 molecules cm-2? As it is now, it seems badly chosen for OMI.

---

## Author Comment (AC1) · 20 Jul 2020

We thank the reviewer for thorough and thoughtful comments. Please see the attached zip file for our responses, the revised manuscript, and the supplemental materials.

Please also note the supplement to this comment: https://www.atmos-chem-phys-discuss.net/acp-2019-1117/acp-2019-1117-AC1-supplement.zip

---

## Author Comment (AC2) · 20 Jul 2020

We thank the reviewer for thorough and thoughtful comments. Please see the attached zip file under AC1 for our responses, the revised manuscript, and the supplemental materials.

---

## Author Response (AR1)

We thank the two reviewers for their thoughtful comments. Our responses are in blue. Trackable changes relative to the last manuscript are also attached at the end.

**Anonymous Referee #1**

In this paper, Zhu et al. evaluated OMI HCHO retrievals using in situ measurements from 12 airborne field campaigns (mostly over the U.S.) and the GEOS-Chem chemical transport model. They first compared GEOS-Chem simulated HCHO with aircraft measurements and used a constant scaling factor for each field campaign to correct the biases in GEOS-Chem simulations. The bias-corrected GEOS-Chem HCHO vertical column densities (VCDs) were then compared with OMI retrievals. It was found that OMI retrievals were generally biased high for low-HCHO conditions and had relatively small biases for high-HCHO cases. Potential reasons for the biases in OMI retrievals were also discussed. Overall, this is a useful study for understanding the quality and uncertainty in OMI HCHO retrievals. The authors expanded upon their previous study using the same approach (Zhu et al., 2016) to different locations and seasons. The advantage of the approach is that by using GEOS-Chem as a transfer standard, data from different field campaigns (often involving different instruments) can be used in a more consistent way for satellite validation. The paper is generally well written. There are, however, several points that need to be clarified. I would recommend that the paper be accepted for publication after the following comments have been addressed.

Specific Comments:

González Abad et al. (2015) showed that there were fairly large changes in OMI HCHO retrievals over remote background areas between the beginning of OMI (2004-2005) and more recent years (2012 and later). All the field campaigns discussed in this study took place after 2013. Have the authors looked into data from earlier field campaigns to check, for example, whether OMI retrieval biases under low-HCHO conditions were smaller in earlier years?

To our knowledge, there were only two flight campaigns, INTEX-B (2006) and MILAGRO (2006), before 2013 that reported in situ HCHO observations. Unfortunately, we failed to obtain INTEX-B HCHO data, and MILAGRO flight campaign is too localized over the Mexico City, so neither one was included in this validation study. The difference in HCHO retrievals over remote background regions between the beginning of OMI and more recent years is mainly due to the aging of the OMI instrument, which was further examined by analyzing the drift of HCHO columns over the remote Pacific [Zhu et al., 2014; 2017a]. OMI HCHO retrieval group at Harvard-SAO has been aware of the issue, and will fix it in the next version SAO HCHO product. We have added one sentence in the text to address the reviewer's concern, please see line 87-88.

Given the positive biases over low-HCHO areas and (smaller but mostly negative) biases over high HCHO areas in GEOS-Chem simulations, using a fixed scaling factor may not be appropriate for some of the field campaigns. For example, as shown in Figure 4, the mean HCHO VCD can range from ~ 3*10^15 molecules/cm^2 to over 10^16 molecules/cm^2 and it is possible that GEOS-Chem overestimates HCHO over relatively clean areas in the southwest corner of the

domain, and at the mean time underestimates HCHO over the southeast corner of the domain. Also by changing the domain slightly (for example, expand the domain to the west), the comparisons between OMI and bias-corrected GEOS-Chem can probably yield somewhat different results.

Accepted. We now limit the validation results (*e.g.*, in Table 2) only to the grids sampled by the aircraft (marked with open circles in Figure 4 and S1-S11).The manuscript is updated accordingly.

The authors proposed the validation method as a "global validation platform". Since the field campaigns discussed in this paper had no coverage outside of North America and Asia Pacific region, this is hardly "global".

Accepted. We have deleted the word "global".

Also for the method to be a "platform", there should be some utilities or applications that facilitate the validation of not just NASA OMI HCHO retrievals, but also retrievals from other satellite sensors/algorithms. Maybe there should also be functionality to ingest additional field campaign data and/or CTM simulations to be used in validation. I wonder if the authors can discuss these aspects of the validation platform (probably as an appendix).

Accepted. We now have added Appendix B to further discuss how to ingest additional field campaign and CTM data. Please see line 325-342.

Technical Comments:
Figure 3 is discussed quite extensively in two parts: lines 119-125 and lines 155-161.Maybe the authors can re-organize and consolidate the two parts?

We appreciate the reviewer's comment. We tried to change the text structure by re-organizing the two parts. However, we may want to stick to the current layout as it fits better to the logic flow: lines 119-125 (Section 2) are about the observed HCHO vertical profiles, while lines 155-161 (Section 3) are about GEOS-Chem modeled results. Figure 3 shows both observed and modeled HCHO vertical profiles, which is an essential part of this study, so that it may deserve in-depth discussion.

**Anonymous Referee #2**

Zhu et al. present a new validation platform for satellite HCHO products, using different aircraft measurements and the model GEOS-Chem as the inter-comparison method. The model is used to make the link between the localised aircraft vertical measurements and the global satellite vertical columns. The method was introduced by Zhu et al. (2016). It is now extended to a larger number of aircraft campaigns, covering a broad range of conditions. An extensive evaluation of the NASA operational OMI HCHO satellite product is presented as an application of the platform. The retrieval steps of the satellite product are examined separately in order to explain the differences between satellite and aircraft results. The paper concludes with a slight bias of the OMI satellite product for high HCHO level, and to a high positive bias for low columns. This study addresses the need for more systematic validation of the satellite products. The use of several aircraft measurements combined with 3D-CTM is pretty new. It allows for direct and indirect validation of the vertical HCHO profiles, which is lacking in the community. A significant amount of work has been dedicated to this goal, and to present the results in an honest and clear way. However, the paper would benefit from some clarifications, especially at the end of the discussion. I recommend publication after the following points have been improved:

The authors claim for a global validation platform. This is supposed to be achieved through the model, but nothing is shown about this extension of the validation beyond the aircraft domains. Can the author add an illustration of this extension? A global map, or a comparison at another location? Alternatively, remain focused on the regions covered by the aircraft campaigns and leave out the reference to the global method.

Accepted. We have deleted the word "global". We may want to point out that a global map is given in Figure 1 (panel a).

In the abstract, it is stated that the high biases are due to slant column fitting and radiance sector correction. It is not clearly demonstrated in the paper.

Accepted. We have rewritten this sentence in the abstract. Please see line 28-29.

The last paragraph of section 4 needs to be revised. The explanations are not clear, and conclusions are drawn without showing any results. It does not hold by itself, and needs to be extended (see my comments below).

Accepted. We have deleted and rewritten related sentences. Please see line 248-253.

The study look for systematic biases: Pay attention to data selection that can also introduce systematic bias (see my comments below).

Accepted. We now use new criteria ($-0.8 \times 10^{16}$ molecules cm$^{-2}$ to $7.6 \times 10^{16}$ molecules cm$^{-2}$) to filter out the outliner, please see line 198-200. The manuscript is updated accordingly.

Does the biases found in this study match the error provided in the satellite product? What about precision? Does the validation results agree with provided satellite product precision?

We appreciate the reviewer's comment. Unfortunately, precision information is not provided in SAO product as reported by [González Abad et al. 2015]. We have added one sentence in the text to address the reviewer's concern. Please see line 198-200.

Introduction p3, line83: "global validation platform . . ... Using observations from 12 aircraft campaigns all over the world". I don't fully agree that this study is global. Results are shown only at the aircraft campaign locations. And those cover mostly US, Pacific and one over Korea. Important emission regions are missing all over the world. I would rather mention here the diversity of the seasonal coverage.

Accepted. We have deleted the word "global".

Application to NASA operational HCHO product Line 169: the so called "radiance reference sector correction term s0"is a slant column. Why do you use the term "radiance"? It is confusing. Does it refer to the reference spectra used during the DOAS fit? If yes, it should be better explained.

Accepted. We have replaced "radiance reference sector correction" with "reference sector correction" in the text.

Line 174: it is said that s0 is the difference between the retrieved SCD over the Pacific Ocean the GEOS-Chem climatology. Please explain if the model vertical columns are directly subtracted from the satellite slant columns (assumption AMF ~ ~ 1 over Pacific Ocean, which is not true), or if an air mass factor is used to convert the GEOS-Chem vertical columns into slant columns.

Accepted. We have rewritten this sentence to clarify. Please see line 177-178.

Line 194: selection criteria (4): VCD within the range $-0.5x10^{16}$ molecules $cm^{-2}$ to $1.0xe10^{17}$ molecules $cm^{-2}$. Please provide the dispersion of the OMI HCHO VCD. If the VCD dispersion is, let's say, 0.8x1016 molecules cm-2, the lower limit might be too strict, removing a significant part of the distribution (negative columns) when the averaged column is close to zero, therefore biasing the averaged column to high values (case"high biases under low-HCHO conditions"). I suggest to test the impact on the comparisons if a more relaxed selection, based on a classical 3approach, is used.

Accepted. We now use new criteria ($-0.8x10^{16}$ molecules $cm^{-2}$ to $7.6x10^{16}$ molecules $cm^{-2}$) to filter out the outliner, please see line 198-200. The manuscript is updated accordingly.

Line 207: Is the limit of $1.1 \times 10^{16}$ for high-HCHO conditions based on satellite or model results? It looks like it is based on satellite columns, which is strange since it is the dataset to be validated, and it presents biases.

Accepted. The limit is based on satellite results. We have rewritten this sentence to clarify. Please see line 213-215.

Line 213: (2) please elaborate on the differences between the "radiance reference sector correction". Same for point (3) "selection criteria".

Accepted. We have removed point (2), and rewritten point (3). Please see line 221.

Line 223: Half of the bias in high-HCHO conditions can be attributed to a priori profiles but not the full bias. Please discuss other possible reasons for the remaining bias.

Accepted. We have rewritten related sentences to clarify. Please see line 225-235.

Line 229: Second case, low-HCHO conditions: I don't agree with the first conclusion. The a priori profiles are not the unique error source in the AMF calculation. It is not because the use of a more precise profile does not improve the comparison that the error is not due to AMF uncertainties. It could be due to albedo uncertainty; or imperfect cloud correction. However, I agree with the rest of the paragraph. Please reformulate the second line.

Accepted. We have rewritten this sentence to clarify. Please see line 237-238.

Line 240: "The SAO retrieval algorithm conducts the radiance reference sector correction by removing the contribution of HCHO over the remote Pacific Ocean to the radiance reference." What are the units of this contribution? A slant column or a radiance? Please explain what you mean by radiance reference. Line 241: Could you give a number for this HCHO contribution? Line 245: What do you mean by "suppressing removal of HCHO contribution in the radiance reference"? Line 246: "the mean bias is reduced from 147% to 128%". Where does this 128% come from? Furthermore, the bias reduction is not significant.

Accepted. We have removed this part to avoid ambiguity.

Line 248: "we attribute the remaining biases to (2) the latitudinal dependency of the radiance reference sector correction." Again this needs further explanations. Is there a latitudinal dependency of the slant columns, if yes positive or negative? And is there a latitudinal dependency in the current reference sector correction of the SAO product?

Accepted. We have removed this part to avoid ambiguity.

Line 249: You find a significant correlation of the satellite HCHO columns with the surface albedo during some campaigns, maybe even larger than the correlation with the model columns. It is interesting. Please specify for which campaigns?

Accepted. The campaigns are now listed in the text. Please see line 250.

If the reflectance climatology contains uncertainties, it will reflect in AMF uncertainties, that will not correct the slant columns for this kind of dependency. Therefore, it is important to know if AMF are used to compute s0. See my comment for line 174. Also, have you tested to remove sun glint scenes from the comparison for the Pacific regions?

Accepted. We have rewritten this sentence to clarify, please see line 177-178. Based on our understanding, sun-glint scenes were already removed in the retrieval process sing the sun-glint warning flag.

Figure 4 and supplement: spatial correlation between model and satellite column seems rather low. Please add correlations in Table 2.

Accepted. Spatial correlations between model and satellite columns are now given in Table 2.

Why not using the same scale for OMI and the model? for example from 0 to $20 \times 10^{15}$ molecules $cm^{-2}$? As it is now, it seems badly chosen for OMI.

Accepted. We now use the same scale for OMI and modeled columns.

[revised manuscript text omitted]